# Capsules with Ileocolonic-Targeted Release of Vitamin B_2_, B_3_, and C (ColoVit) Intended for Optimization of Gut Health: Development and Validation of the Production Process

**DOI:** 10.3390/pharmaceutics15051354

**Published:** 2023-04-28

**Authors:** Aisha A. Ahmed, Antonius T. Otten, Bahez Gareb, Judith E. Huijmans, Anko C. Eissens, Ateequr Rehman, Gerard Dijkstra, Jos G. W. Kosterink, Henderik W. Frijlink, Reinout C. A. Schellekens

**Affiliations:** 1Apotheek A15, 4207 HT Gorinchem, The Netherlands; a.a.ahmed@lumc.nl (A.A.A.);; 2Department of Gastroenterology and Hepatology, University Medical Center Groningen, University of Groningen, 9713 GZ Groningen, The Netherlands; 3Department of Clinical Pharmacy and Pharmacology, University Medical Center Groningen, University of Groningen, 9713 GZ Groningen, The Netherlands; 4Department of Pharmaceutical Technology and Biopharmacy, Groningen Research Institute of Pharmacy, University of Groningen, 9713 AV Groningen, The Netherlands; 5DSM Nutritional Products, 4303 Kaiseraugst, Switzerland; 6Department of Pharmaco Therapy, Epidemiology and Economics, Groningen Research Institute of Pharmacy, University of Groningen, 9713 AV Groningen, The Netherlands

**Keywords:** drug targeting, vitamin B_2_, vitamin B_3_, vitamin C, ileocolonic delivery, antioxidant, microbiome, inflammatory bowel disease, development, validation, production

## Abstract

The ileocolonic-targeted delivery of vitamins can establish beneficial alterations in gut microbial composition. Here, we describe the development of capsules containing riboflavin, nicotinic acid, and ascorbic acid covered with a pH-sensitive coating (ColoVit) to establish site-specific release in the ileocolon. Ingredient properties (particle size distribution, morphology) relevant for formulation and product quality were determined. Capsule content and the in vitro release behaviour were determined using a HPLC-method. Uncoated and coated validation batches were produced. Release characteristics were evaluated using a gastro-intestinal simulation system. All capsules met the required specifications. The contents of the ingredients were in the 90.0–120.0% range, and uniformity requirements were met. In the dissolution test a lag-time in drug release of 277–283 min was found, which meets requirements for ileocolonic release. The release itself is immediate as shown by dissolution of the vitamins of more than 75% in 1 h. The production process of the ColoVit formulation was validated and reproducible, it was shown that the vitamin blend was stable during the production process and in the finished coated product. The ColoVit is intended as an innovative treatment approach for beneficial microbiome modulation and optimization of gut health.

## 1. Introduction

Imbalances in gastrointestinal health and specifically disruptions in human gut microbiota homeostasis are associated with a vast variety of health disorders, including inflammatory, metabolic, and psychological disorders [1,2,3]. Gut microbiome–host interactions are linked to pathophysiological aspects of different types of diseases, but also deemed predictive of treatment response and disease progression [4,5]. Microbiota manipulation has thus become of great scientific interest, as controlled modification of microbiota composition is expected to influence disease onset and optimize treatment outcomes. Microbiota modulation through nutrition is of particular interest, as a variety of nutritional components and micronutrients have showcased shifts in gut microbiota composition that promote beneficial changes in gut health status [6,7,8].

Vitamins are essential micronutrients for humans and are involved in numerous biological processes. Vitamins support the immune system and function as essential precursors for enzymes involved in cellular energy generation [9]. Supplementation with vitamins delivered to the gut microbiota residing in the ileocolonic regions, especially the antioxidant B- and C-vitamins, has recently been identified as a promising treatment strategy to establish potentially beneficial alterations in gut microbial composition and function [10]. Through high dose supplementation and targeted colonic delivery, B- and C-vitamins have shown the ability to affect the relative presence of specific bacterial strains, leading to an increased presence of health-promoting metabolites, such as short chain fatty acids (SCFAs), in the gut lumen [11,12,13,14].

An oral ileocolonic-targeted formulation of vitamins circumvents the efficient vitamin-uptake in the upper gastrointestinal tract. A targeted approach establishes a controlled, site-specific, and convenient exposure of the intestinal luminal environment, and subsequently the microbiota residing in the ileocolonic regions, to vitamins [15]. Oral (ileo)colonic drug delivery can be accomplished by exploiting various site-specific features and stimuli naturally occurring in the gastro-intestinal tract, such as intestinal transit time, pH-profiles, and gut microbial-derived enzymes [16,17,18]. Oral pH-mediated drug delivery is a reliable targeting strategy, as intestinal pH-profiles, especially the prominent peak in pH of ≥7 in the ileum, do not display substantial interindividual fluctuations nor does intestinal disease (i.e., a state of inflammation) have a detrimental effect on ileal intraluminal pH [19]. Gastro-intestinal transit times are intrinsically more variable, and they can be substantially altered by various disease states, evidenced by patients with inflammatory bowel disease (IBD) with intestinal inflammation. In this group of patients, small intestinal transit time is often prolonged, while colonic transit time is accelerated during active inflammation, possibly due to the occurrence of diarrhoea [20,21]. Accelerated transit times can potentially lead to suboptimal drug release of pH-sensitive targeted oral formulations. Existing commercially available oral ileocolonic targeted formulations display signs of suboptimal release characteristics (i.e., untimely release before the ileocolonic segments is reached, incomplete drug release during transit through the colon), suggestive of sub-optimal efficacy of the compounds [22,23,24,25]. Optimization of established oral targeted delivery techniques can thus increase efficacy of pharmaceutical agents on a short notice.

The ColoPulse is a coating technology which offers optimized drug exposure facilitating pulsatile drug release. The ColoPulse leverages a super disintegrant (e.g., Ac-Di-Sol^®^) incorporated into a pH-sensitive polymer (e.g., Eudragit^®^ S) in a non-percolating fashion [26]. Once the pH-threshold of 7 is passed, which corresponds to the physiological spike of intraluminal pH of 7.5 in the terminal ileum [27,28], the outer layer of the coating will start to dissolve and disintegrant will swell because of its interaction with the aqueous intestinal fluids, leading to accelerated rupture of the polymer-based coating and subsequent pulsatile drug release will be accomplished within 60 min following the short exposure to a pH > 7. The intended ileocolonic targeted release of the encapsulated drug from ColoPulse-based products was validated in studies involving both healthy volunteers and patients with gastro-intestinal disease [29,30,31].

In this study, we aim to develop and validate the production process of ColoVit capsules and matching placebo capsules. The ColoVit capsules are ileocolonic targeted vitamin capsules containing a mixture of riboflavin (B_2_), nicotinic acid (B_3_), and ascorbic acid (C). The ColoVit capsule is developed to explore its microbiome-modulating capacities in a clinical trial setting (ClinicalTrials.gov Identifier: NCT04913467) and, ultimately, the ColoVit is intended as an innovative treatment strategy to improve intestinal health status and positively influence the onset and course of microbiota-associated diseases.

## 2. Materials and Methods

### 2.1. Materials

Riboflavin, nicotinic acid, and ascorbic acid were provided by DSM Nutritional Products AG (DSM Nutritional Products AG, Kaiseraugst, Switzerland). Microcrystalline cellulose (Avicel^®^PH101 and Avicel^®^PH102, DuPont, Paris, France), a mixture of calcium hydrogen phosphate dihydrate, sodium starch glycolate and silicon dioxide (94:5:1) (Fagron, Capelle aan den IJsel, The Netherlands), magnesium stearate (Spruyt Hillen, Capelle aan den IJsel, The Netherlands), silicon dioxide (Spruyt Hillen, Capelle aan den IJsel, The Netherlands), methacrylic acid-methyl methacrylate copolymer 1:2 (Eudragit^®^ S100, Evonik industries AG, Essen, Germany), croscarmellose sodium (Ac-di-Sol^®^, DuPont, Paris, France), talc (Duchefa Farma, Haarlem, The Netherlands), Polyethylene glycol 6000 (VWR, Amsterdam The Netherlands), acetone (Duchefa Farma, Haarlem, The Netherlands), red gelatin capsules size 0 (Spruyt Hillen, Capelle aan den IJsel, The Netherlands), and purified water were used for the production of the capsules. Riboflavin, nicotinic acid, and ascorbic acid were all of food grade, the other ingredients were pharmacopoeia grade (Ph. Eur. or USP/NF).

### 2.2. Formulation of Uncoated Capsules

Raw materials were characterized to investigate optimal composition and formulation of the powder mixture. Particle size distributions were investigated using laser diffraction particle size analyser HELOS/BF (Sympatec GmbH, Clausthal-Zellerfeld, Germany). Particle morphology was visualized by scanning electron microscopy (JEOL JSM-6301-F, Jeol, Akishima, Japan) with a magnification of 50 times. Powder mixtures of the verum were made for performance testing. Overages of the vitamins were used to compensate stability or production losses (Table 1). The composition of the different formulations per capsule is shown in Table 2. Flow properties were assessed by determining the angle of repose (funnel method) and bulk vs. tapped powder density (Hausner ratio). The physical stability of different powder mixtures was determined by examining the particle morphology and homogeneity and by measuring the particle size distribution. The mixtures were placed into a mortar; after tapping in a controlled manner for ten or twenty times, respectively, mixture homogeneity was visually determined by assessing the number of white particles segregated at the surface. 

For the placebo capsules a mixture of caffeine, Avicel PH102, magnesium stearate, and silicon dioxide were further mixed with calcium hydrogen phosphate dihydrate, sodium starch glycolate, and silicon dioxide (94:5:1). This material was used as the capsule filling powder. The exact composition of the placebo capsule is given in Table 2. Capsules were filled on an automated filling capsulating machine (INCAP, Dott. Bonapace & C, Cusano Milanino, Italy) operated at a rate between 1500 and 3000 capsules per hour with a target fill weight of 425 mg. 

### 2.3. Production and Quality of Uncoated Capsules

Uncoated capsules were produced by a 3-step process comprising the unit operations weighing of ingredients, mixing of the ingredients with a cube mixer (Indola Holland, Rijswijk, The Netherlands), and subsequent automated capsule filling (IN-CAP, Dott. Bonapace & C, Cusano Milanino, Italy). Batch size was determined at 3.02 kg equivalent to a gross yield of 7100 units. Quality requirements were based on pharmacopeial monographs: Ph. Eur. Dosage forms, Capsules (0016), Ph. Eur. 2.9.1, Ph. Eur. 2.9.3, and Ph. Eur. 2.9.40 in combination with in-house specifications (Table 3). 

### 2.4. Process Validation

The process validation included both the mixing process and the capsule filling process. The mixing process of the verum was validated by mixing riboflavin, ascorbic acid, nicotinic acid, caffeine, Avicel^®^PH101, and silicon dioxide for 30 min at 41 RPM in the cube mixer. For the placebo caffeine, Avicel^®^PH102, silicon dioxide and a mixture of calcium hydrogen phosphate dihydrate, sodium starch glycolate, and silicon dioxide (94:5:1), were mixed for 30 min at 41 RPM in the cube mixer. Subsequently, magnesium stearate was added to the cube mixer and all the excipients were mixed for another 5 min at 41 RPM. Then powder samples of approximately 425 mg (300–550 mg) were taken in duplicate at the top (T), in the middle (M) and at the bottom of the mixture (B). The contents of caffeine (placebo) and riboflavin, caffeine, and nicotinic acid (verum) were analysed in these samples using a high-performance liquid chromatography (HPLC)-method. 

Capsules were filled with approximately 425 mg of powder mixture per capsule. Samples were taken per 1000 capsules throughout the encapsulating process. Additionally, at the beginning (first 1000 capsules), the middle (capsules 3000–4000) and the end (capsule 6000–7000) of the filling process samples were taken per 200 capsules. Per timepoint the contents of 10 individual capsules were determined and uniformity of dosage units (Acceptance Value) was calculated.

### 2.5. Production and Quality of Coated Capsules

The capsules were coated by a spray coating process using the ProCept 4m8-TriX Pan Coater (ProCept, Zelzate, Belgium) with a drum size of 5 litres. This coater is a so-called perforated drum coater. The nozzle was a Schlick Atomizing Nozzle 1.0 mm. Batch size was 3000 or 4500 capsules. Process parameters and conditions were investigated during a pre-validation phase and during the technology transfer phase; the process parameters are shown in Table 4. The results of the technology transfer are described in Section 3.4 Technology transfer lab scale coater to ProCept Pan Coater. The coat fluid was prepared as previously described [16]: Macrogol 6000 1% (1 g), Eudragit S100 7% (7 g). AcDiSol 3% (3 g), Talc 2% (2 g) in 97 mL acetone + 3 mL purified water. Quality requirements were based on in-house process control specifications of Apotheek A15 and on previous ColoPulse studies (Table 5).

### 2.6. Technology Transfer

During the technology transfer the capsules were coated on two different scales with different coating techniques. The release profile of the capsules was determined by measuring the released caffeine from the coated capsules. The caffeine was analyzed in-line with UV-VIS (Thermo Fisher, Madison, WI, USA) at a wavelength of λ = 272 nm (path length of 10.00 mm) every 3 min for 6 h.

### 2.7. Assay

The HPLC analytical assay of caffeine, riboflavin, nicotinic acid, and ascorbic acid was validated conform the ICH guideline Q2. The required validation characteristics were determined and show excellent method performance (Table 6). The analysis was carried out on a Shimadzu LC20-AD prominence HPLC with a UV-VIS detector. A Symmetry C18 5.0 μm 4.6 × 250 mm column was used. The eluent composition was acetonitrile: phosphate buffer 10 mM pH 5.2 (10:90). The following parameters were used: flow: 1.0 mL/min, injection volume: 10 μL, wavelength: 262 nm, oven temperature: 20 °C, and autosampler temperature: 4 °C. An example of the obtained chromatogram is shown in Appendix A.

### 2.8. In Vitro Dissolution Testing

The release characteristics of various formulated coated tablets were evaluated in a previously described modified dissolution test, the so-called gastro-intestinal simulation system (GISS). The GISS is a 6-hour dissolution test which is based on the pharmacopoeial paddle method (apparatus II, Prolabo, Rhône-Poulenc, Paris, France). During the test the formulation is exposed to four phases. The four phases simulate the the pH-gradient of the gastro-intestinal-tract (stomach-jejunum-ileum-colon), At the end of each phase a switch solution was added to obtain the required composition of the next phase. The pH in the simulated stomach is 1.2, in the simulated jejunum 6.8, in the simulated terminal ileum 7.5 and in the simulated ascending colon 6. The caffeine concentration was measured at four different timepoints: 115 min, 230 min, 300 min, and 360 min. The samples taken at the different timepoints were analysed; the concentration of caffeine was measured with the HPLC-method described in Section 2.7.

The cumulative percentage of the dose released was plotted against time. The release profile was characterized (n = 6) by three outcome parameters. First, the starting point of release is defined as the time point at which the arithmetic mean of the cumulative percentage of the dose released is 5% (lag time). The lag time should exceed 230 min, being the moment that the capsules leave the simulated jejunum. Second, the release rate is reflected by the pulse time, defined as the time between lag time and t70%. The pulse time should preferably be less than 60 min (the pharmacopeial requirement for disintegration of gastro-resistant tablets and capsules). Third, the cumulative drug release at the end (R_360 min_) of the dissolution test should exceed 80%.

## 3. Results

### 3.1. Powder Mixture Formulation

The particle size distribution profiles of riboflavin, nicotinic acid, ascorbic acid, and the excipients used are shown in Table 7. Riboflavin and ascorbic acid share a similar particle size, whereas nicotinic acid is considerably smaller in combination with a much broader span. The electron microscopy images in Figure 1 show the different particle morphologies, with riboflavin consisting of rounded particles as opposed to the angular shapes of both ascorbic acid and nicotinic acid.

Powder mixture performance of the formulations described in Table 2 is shown in Table 8. The homogeneity of the powder mixtures is also visualized in Figure 2. The powder mixtures have only slightly different properties. Formulation 1 has somewhat poorer flow properties, which makes it more stable providing better homogeneity after tapping. For this reason, formulation 1 was used to produce the clinical batches. 

### 3.2. Process Validation

The validation of both the mixing process and the capsule filling process was successfully executed. The results of the process validation are shown in Table 9. During the mixing process, the content of riboflavin and nicotinic acid is over 110%. During the capsule filling process, production losses slightly reduce the content of the vitamins, but they remain within the 90–120% requirement. Additionally, the process validation of the capsule filling process of placebo capsules was likewise executed successfully. The results of this validation with a verum and placebo batch are shown in Table 10.

### 3.3. Production and Quality Control of Uncoated Capsules

The production of the uncoated capsules was executed successfully. All capsules met pharmacopoeial requirements and in-house specifications. At batch release testing, the contents of active ingredients caffeine, riboflavin, ascorbic acid, and nicotinic acid was compliant with the pre-set specifications of 90.0–120.0%. The AV of all the active ingredients was lower than 15.0, the disintegration time was not more than 15 min, and the dissolution was more than 75% in 1 h. There were no remarkable changes observed in the characteristics of the capsules during a storage period of 6 months. The product characteristics of uncoated capsules are shown in Table 10.

### 3.4. Technology Transfer Lab Scale Coater to a Perforated Drum Coater

The coating process described in this paper is a scale-up of a process originally designed at lab scale. The coating process was transferred from a lab scale process with 30 units to a controlled production scale process with 4500 units. In the lab scale setting 30 size 0 capsules were coated with the coating suspension which was continuously sprayed with a Schlick Atomizing Nozzle 1.0 mm onto the capsules in a mini-rotating vessel equipped with a hot air blower for the evaporation of the solvent mixture and to induce film formation. It was not possible to control the process parameters of nozzle gas, pattern gas and inlet gas flow. Only the product temperature could be controlled. To obtain the release pattern that met the quality requirements described in Table 5 the capsules were coated with a theoretical coat thickness between 9.0 and 11.0 mg/cm^2^.

The scaled-up coating process with the ProCept Pan Coater, which is a perforated drum coater, was designed based on the lab scale coating process. With the ProCept Pan Coater process parameters are better controlled (Table 4). With controlling the nozzle gas flow, it was possible to control the coating porosity and with the pattern gas it was possible to control the width of the coating spray. Initially, the capsules were coated with the same theoretical coat thickness as the capsules produced at lab scale; however, during the GISS-dissolution the capsules did not open. Subsequently, the capsules were coated with a theoretical coat thickness of 7.0–8.5 mg/cm^2^ and the release pattern of the ProCept Pan Coater coated capsules was similar to the release pattern of the lab scale coated capsules. The release profile found with capsules from the two coat processes (having different theoretical coat thicknesses) is shown in Figure 3. Both release profiles meet the quality requirements described in Table 5. The product and process characteristics of the capsules of the lab scale coater and ProCept Pan Coater are shown in Table 11.

### 3.5. Production and Performance of Coated Capsules

The production of the coated capsules was also carried out successfully. All capsules met the in-house specifications (Table 3). Due to the maximum load of the drum of the coating machine, the second clinical batch of the uncoated capsules of both verum and placebo (CB.V2 and CB.P2) was divided into two sub-batches for the coating process (CB.V2.1/CB.V2.2 and CB.P2.1/CB.P2.2). The product and process characteristics of the coated batches are shown in Table 12. 

The GISS was used to investigate the coating performance of the ColoVit capsules. The lag-time for all tested batches in the GISS was between 277–283 min. This corresponds to the simulated terminal ileum. After the coating disintegrated, the release of caffeine from the capsules into the simulated lumen was fast and complete.

## 4. Discussion

The ColoVit, an ileocolonic-targeted capsule containing 37.5 mg riboflavin, 2.5 mg nicotinic acid, and 250 mg ascorbic acid, is a novel formulation intended for microbiome-modulation and optimization of gut health. The ColoVit utilizes a pH-sensitive coating technology, the ColoPulse, which enables a pulsatile targeted delivery of the capsule’s content in the terminal ileum and colon [26,29,30,31]. 

Analysis of vitamin content, before and after the application of the coating, showed that the vitamin mixture in the ColoVit preparation was stable during production and subsequent storage. The intended ileocolonic release pattern was confirmed in an in vitro dissolution test simulating the passage through the gastrointestinal tract. The formulation complied with the requirements for oral dosage forms intended for human use stated in the monographs of the European Pharmacopoeia. In the current paper, we present a structured, validated, and reproducible process to produce ileocolonic-targeted vitamin capsules.

To the best of our knowledge this is the first manuscript to describe a validated and reproducible method to develop and produce ileocolonic-targeted vitamin capsules, intended to target the gut microbiome. There is a demand for ileocolonic-targeted formulations with optimized release profiles to reliably deliver pharmaceutical compounds or supplements to the target location, a release profile which would ideally not be affected by an inflammatory state of the intestines or inter- and intraindividual differences of intestinal physiology and function. The ColoVit utilizes an optimized pH-sensitive delivery strategy, and the large-scale production of this formulation has been validated and has proven reproducible. 

Additionally, this manuscript serves as a tech transfer of the coating process from a limited lab scale to larger scale production of clinical batches in a compounding facility. For the development of a new formulation, it is important set up a robust production process. During this study we have shown that we have set up a reproducible and robust production process to produce the uncoated and coated vitamin capsules. We started with the formulation of the uncoated capsules. The verum capsules contain 9.5% riboflavin, 0.68% nicotinic acid, and 63.5% ascorbic acid per capsule. Additionally, 5.9% of caffeine is added to the capsules as a marker substance for the drug release. Caffeine is added as a marker substance, because compared to the vitamins it is stable in the different dissolution media of the gastro-intestinal simulation system (GISS) and it has similar dissolution properties.).

Combining four different compounds in a single dose brings different challenges. The difference in particle size, particle morphology, density, and quantity of the four compounds are critical for the homogeneity of the powder mixtures and the content uniformity of the capsules. To successfully combine the compounds in a single dose we have characterized all the raw materials and formulated three different powder mixtures. The powder mixture with Avicel^®^PH101 (formulation 1, as described in Table 2) had a slightly poorer flowability than the powder mixtures with Avicel^®^PH102 (formulation 2) and a mixture of Avicel^®^PH101: Avicel^®^PH102 (50:50) (formulation 3). However, the visual homogeneity of formulation 1 was better than the other two formulations. This can be explained by the particle size of Avicel^®^PH101; Avicel^®^PH101 has a smaller nominal particle size than Avicel^®^PH102. The smaller nominal particle size fits better with the different particle sizes of the four compounds, which results in a more stable formulation. On top of that, the slightly poorer flowability also results in a more stable formulation without causing problems during the filling process.

The initial formulation was based on applying an overage of 15% nicotinic acid and 8% riboflavin, to compensate for losses during production and storage. However, it was found that the production loss of nicotinic acid was lower than 10%, indicating that the production overage of nicotinic acid could be lowered to 5%. The same could have been carries out for the stability overage of riboflavin. For riboflavin, a stability overage of 8% was added; however, during storage and production the content of riboflavin hardly decreases. Therefore, an overage of 5% would be more suitable for follow-up batches.

After the development and validation of the uncoated capsules, the development and the validation of the coating process started. The coating process was transferred from a lab scale process (30 units, coating duration of approximately 30–60 min) to a controlled production scale process (4500 units, coating duration of approximately 4 h) [32,33]. Unfortunately, we were not able to control and measure all the coating parameters of the lab scale coater. The only parameters that could be measured were the drum speed, the pump speed, and the temperature. Experience with the lab coater has shown that during the coat process the temperature of the product has to be between 25 °C and 30 °C to obtain the desired properties of the coating. Variation in the drum and pump speed only influenced the total coating duration, but it did not affect the properties of the coating. However, the pump speed was set up to the highest possible speed on the pump to prevent clogging.

Interestingly the different equipment and process parameters allowed us to apply a 25% thinner coating (8 vs. 10 mg/cm^2^) while still obtaining similar release properties. This difference is explained by a higher density of the coating as produced at production scale. The coating density is the volume of coating material used to cover the area of the capsules. The reasons for this are the higher forces exerted on the coating due to the increased mass and the longer duration of the process. Additionally, we were able to control the nozzle gas which was not possible at lab scale. Increasing the nozzle gas results in the formation of smaller particles, this eventually leads to a higher coating density (and reduced porosity of the coating). The porosity of the coating can be described by the area of the capsule that is not covered with the coating: in other words, the holes in the coating. The prolonged processing time, the formation of smaller particles, and the stability of the coat process with the ProCept coater also led to a more homogeneous coverage of the capsules with the coating. Due to the prolonged processing time the polymer particles have more time to coalesce, which results in a more homogeneous coating. Conversely, with the lab scale coater a higher coating solution mass is required to achieve the same homogeneous coverage. These phenomena explain the similarity between the release profiles of products from the different processes despite the difference in theoretical coat thickness. 

The dosage of 37.5 mg riboflavin, 2.5 mg nicotinic acid, and 250 mg ascorbic acid per capsule is based on the dosages used in previous studies which reported beneficial microbiota-modulation and potential health-promoting effects [11,12,13,14,34,35,36]. In healthy subjects, 1000 mg/daily conventional- and 500 mg/daily colon-targeted ascorbic acid supplementation have demonstrated shifts in gut bacterial populations and increased presence of favourable microbiota-derived metabolites [11,12] 50–100 mg/daily conventional- and 75 mg/daily colon-targeted riboflavin supplementation altered gut microbial composition and function [12,13]. Likewise, 30–300 mg/daily pH-sensitive delayed release nicotinic acid formulation introduced shifts in gut bacterial populations [36]. Effects on gut microbes appeared most prominent when riboflavin and ascorbic acid when given in a colonic-targeted combination formulation of both vitamins [12]. Additionally, the proposed vitamin dosage per capsule is below the tolerable daily upper intake levels for adults at which adverse effects can occur, as set by food safety Authorities such as the US Council for Responsible Nutrition and Scientific Committee on Food of the European Food Safety Authority [37]. 

Previous investigations concerning coating performance of the ColoPulse show that the coating is stable for 12 months storage at room temperature [32,33]. Therefore, it is expected that the stability of the vitamins, the ileo-colonic targeting performance as well as the drugs release characteristics of the ColoVit capsules will be maintained during the storage period of 12 months stored at room temperature. The stability of the formulation will be tested during a period of two years.

Mechanisms through which vitamins can impact gut health and modulate gut microbiota are suspected to be both direct, by influencing metabolic processes of commensal gut bacteria, and indirect by introducing improvements in luminal ileocolonic conditions which allow for increased growth and activity of specific microbial clusters [10].

B- and C-vitamins are intrinsically produced by specific gut microbial strains and cross-feeding to other microbiota occurs [38,39]. Commensal microbiota residing in the intestines use B- and C-vitamins as support for different cellular processes. For example, *Faecalibacterium prausnitzii*, a gut bacterial species often associated with health-promoting processes, uses riboflavin for growth and utilizes the vitamin to increase the anaerobe microbe’s endurance in oxygenated, potentially harmful, environments [40].

As essential co-factors, B- and C-vitamins directly support gut immunity and epithelial barrier function [41,42]. Additionally, B- and C-vitamins are considered potent antioxidants. Activated leukocytes and macrophages can produce large amounts of reactive oxygen species (ROS) both on a systemic and an intestinal localized level, predisposing to oxidative stress disturbances and DNA damage [43,44]. Targeted antioxidant vitamin supplementation can actively scavenge ROS in the intestinal lumen, which mitigates oxidative stress, and thus moderates tissue damage and inhibits inflammation.

Clinical studies investigating riboflavin, nicotinic acid, and ascorbic acid have demonstrated an increase in microbial diversity after supplementation and an increased presence of SCFAs in the intestinal lumen [11,12,13,14,34,35,36]. SCFAs, such as acetate, propionate, and butyrate, are critical gut microbiota-derived metabolites that preserve integrity of the intestinal barrier and function as energy source for colonocytes, and therefore play an important role in maintaining intestinal homeostasis [45,46]. Furthermore, targeted riboflavin and ascorbic acid supplementation have both showcased potential favourable changes in microbial composition through reduction in Enterobacteriaceae [11,35]. Enterobacteriaceae are generally assumed to be pathogenic, as this bacterial family has been associated with enhanced inflammatory responses and overgrowth of Enterobacteriaceae contributes to increased oxygen concentrations in the intestinal lumen, which actively suppresses the survival of other microbial strains [47].

As disruptions in gut health and microbiota homeostasis are linked to a variety of diseases [1,2,3], the ColoVit might be of future interest for a broad range of conditions. For example, antibiotic use-induced decreases in gut microbial diversity are linked to adverse gastrointestinal effects, providing a rationale for preventive strategies to optimize gut health before antibiotics are initiated [48]. Additionally, decreased abundances of SCFA-producing bacteria and thus decreased intestinal presence of SCFA were linked to increased risks of colorectal cancer [49], while high diversity profiles can positively modulate treatment response to cancer immunotherapies, such as immune checkpoint inhibitors [50,51]. Reductions in microbial diversity and SCFA-producing bacterial strains are negatively correlated with onset and disease severity in patients with IBD [52,53]. Considering these strong associations of microbiome dysbiosis with IBD disease activity, taken together with findings that vitamin deficiencies are common in this specific group of patients [54,55], and this group is generally exposed to elevated levels of ROS [56], targeted suppletion of antioxidant vitamins can be considered a promising treatment strategy. 

From a regulatory perspective, it is not yet clear were to position the ColoVit capsule. It can be regarded as a pharmaceutical product or a s a dietary supplement with additional proposed health-promoting properties that might go beyond fundamental nutritional values of vitamins. In this context, the ColoVit resides in the grey area between a dietary supplement and a pharmaceutical product, a phenomenon sometimes referred to as a nutraceutical [57,58]. The categorization and subsequent regulations of such products remains up for debate [59]. 

For our clinical trial purpose, ColoVit capsules were classified as a food supplement, thus different criteria apply regarding product quality as compared to a pharmaceutical product while always maintaining efficacy and safety of the product. Firstly, a broader specification for the content of the vitamins was applied at end of shelf-life at 80–120% instead of the pharmaceutical product limit of 90–110%. Secondly, while controlling the content of each vitamin, the possible development, qualification, and quantification of degradation products is not required for food products.

## 5. Conclusions

The ColoVit is a capsule containing a vitamin B_2_, B_3_, and C mixture, with a targeted release of the vitamins to the ileocolonic region, utilizing the pH-sensitive ColoPulse coating technology. The product is intended as an innovative treatment approach for beneficial microbiome modulation and optimization of gut health. The production process of this formulation was validated and found to be reproducible. Riboflavin, nicotinic acid, and ascorbic acid were stable in uncoated and coated capsules. Therefore, the ColoVit is an interesting novel formulation for the oral treatment of diseases associated with disruptions in gut microbiota composition and function. A double blinded, randomized clinical trial investigating the effects of the ColoVit on gut microbial communities and disease course of patients with Crohn’s disease and healthy volunteers is currently being conducted [60] (ClinicalTrials.gov Identifier: NCT04913467)**.**

## 6. Patents

The ColoPulse coating technology is patented [61].

## Figures and Tables

**Figure 1 pharmaceutics-15-01354-f001:**
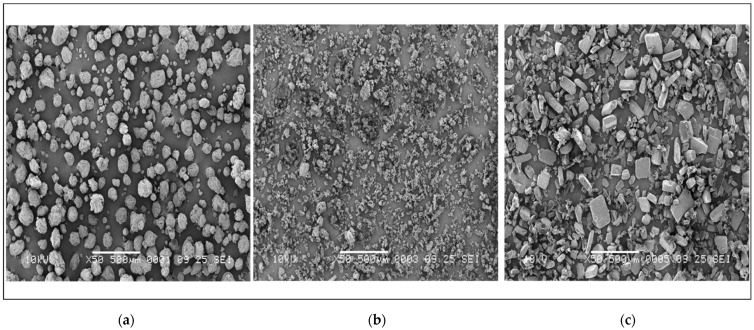
SEM-images of (**a**) riboflavin, (**b**) nicotinic acid and (**c**) ascorbic acid. The powders have been magnified up to 50 times, the bar on the pictures is equal to 500 µm.

**Figure 2 pharmaceutics-15-01354-f002:**
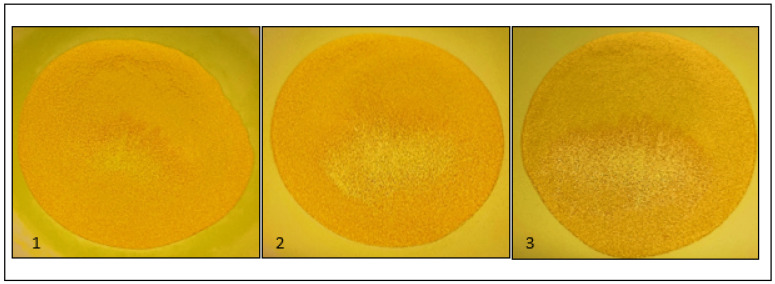
Visualization of the three different formulations after tapping 10 times. The homogeneity is determined based on the number of white particles at the powder surface. 1: formulation 1, 2: formulation 2 and 3: formulation 3 as shown in Table 2.

**Figure 3 pharmaceutics-15-01354-f003:**
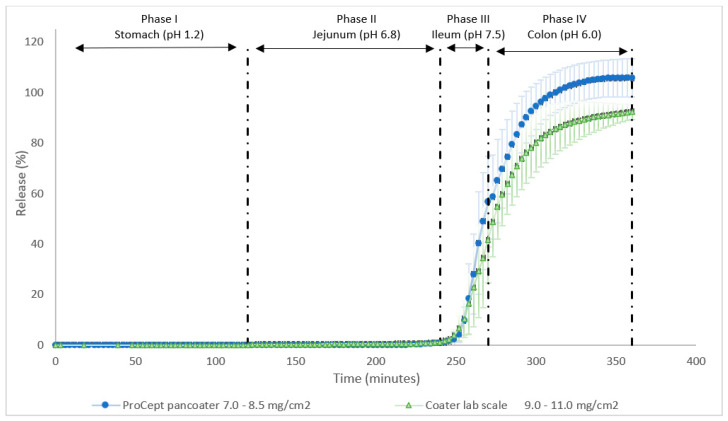
Release profile of caffeine of the lab scale coated capsules (green triangles) and ProCept Pan Coater coated capsules (blue circles) during a 6-h GISS.

**Table 1 pharmaceutics-15-01354-t001:** Vitamin composition of the ColoVit capsules.

	Label Claim	Production Overage	Stability Overage	Total
Riboflavin	37.5 mg (100%)	1.125 mg (3%)	1.875 mg (5%)	40.5 mg (108%)
Nicotinic acid	2.5 mg (100%)	0.25 mg (10%)	0.125 mg (5%)	2.875 mg (115%)
Ascorbic acid	250 mg (100%)	7.5 mg (3%)	12.5 mg (5%)	270 mg (108%)

**Table 2 pharmaceutics-15-01354-t002:** Composition of three different formulations for the verum capsules and the composition of the placebo capsule.

	Verum			Placebo
	Formulation 1	Formulation 2	Formulation 3	
Riboflavin	40.5 mg (9.5%)	40.5 mg (9.5%)	40.5 mg (9.5%)	----
Nicotinic acid	2.875 mg (0.68%)	2.875 mg (0.68%)	2.875 mg (0.68%)	----
Ascorbic acid	270 mg (63.5%)	270 mg (63.5%)	270 mg (63.5)	----
Caffeine	25 mg (5.9%)	25 mg (5.9%)	25 mg (5.9%)	25 mg (5.9%)
Avicel^®^PH101	80.3 mg (18.9%)	-----	40.1 mg (9.4%)	----
Avicel^®^PH102	----	80.3 mg (18.9%)	40.1 mg (9.4%)	128.6 mg (30.3%)
Magnesium stearate	2.1 mg (0.49%)	2.1 mg (0.49%)	2.1 mg (0.49%)	2.1 mg (0.49%)
Silicon dioxide	4.3 mg (1.0%)	4.3 mg (1.0%)	4.3 mg (1.0%)	1.5 mg (0.35%)
Calcium hydrogen phosphate dihydrate, sodium starch glycolate and silicon dioxide (94:5:1)	----	----	----	267.6 mg (63.0%)
Total weight	425.1 mg (100%)	425.1 mg (100%)	425.0 mg (100%)	424.8 mg (100%)

**Table 3 pharmaceutics-15-01354-t003:** Quality requirements of the uncoated capsules.

Parameter	Specification Release	Specification End-of-Shelf Life	Reference
Appearance	Undamaged, red capsules containing orange powder (verum)	Undamaged, red capsules containing orange powder (verum)	In-house method (visual inspection)
	Undamaged, red capsules containing white to off-white powder (placebo)	Undamaged, red capsules containing white to off-white powder (placebo)	
Identity	Conform reference for caffeine, riboflavin, ascorbic acid and nicotinic acid (verum)	N.A.	In-house method (HPLC)
	Conform reference for caffeine. Riboflavin, ascorbic acid, nicotinic acid not detected (placebo)		
Assay and uniformity of dosage unit caffeine	90.0–110.0%	For information only	Ph. Eur. 2.9.40 Uniformity of dosage units (HPLC: in-house method)
	Acceptance value ≤ 15.0	N.A.	
Assay and uniformity of dosage units riboflavin	90.0–120.0%	80.0–120.0% (verum)	
	Acceptance value ≤ 15.0	N.A.	
Assay and uniformity of dosage units ascorbic acid	90.0–120.0%	80.0–120.0% (verum)	
	Acceptance value ≤ 15.0	N.A.	
Assay and uniformity of dosage units nicotinic acid	90.0–120.0%	80.0–120.0% (verum)	
	Acceptance value ≤ 15.0	N.A.	
Disintegration	NMT 15 min	NMT 15 min	Ph. Eur. 2.9.1. Disintegration of tablets and capsules
Dissolution ^1^	NLT 75% in 1 h	NLT 75% in 1 h	USP <2040> Disintegration and Dissolution of Dietary Supplements, Ph.Eur. 2.9.3 Dissolution test for solid dosage forms

^1^ Verum: measured on riboflavin as index vitamin, placebo: measured on caffeine.

**Table 4 pharmaceutics-15-01354-t004:** Overview of the process parameters for the coating process.

Process Parameter	Value
Coating Inlet gas flow (m^3^/min)	0.40
Inlet temperature (°C)	40
Drum speed (rpm)	15
Nozzle gas (bar)	0.49
Pattern gas (bar)	0.49
Pump speed (rpm)	400
Product temperature (°C)	25–30

**Table 5 pharmaceutics-15-01354-t005:** Quality requirements for the coated capsules.

Parameter	Specification
Appearance	Red capsules with a frosted appearance. Coating has no cracks (visual verification).
Theoretical coat thickness	7.0–8.5 mg/cm^2^
Lag time	Concentration caffeine < 5% at 230 min
Pulse time (T70% (min))	Concentration caffeine > 70% at 300 min
Cumulative release	Concentration caffeine at 360 min (end of test) > 80%

**Table 6 pharmaceutics-15-01354-t006:** Validation characteristics of the HPLC analytical assay method for caffeine, riboflavin, nicotinic acid, and ascorbic acid. The method is validated for accuracy, specificity, linearity, precision, repeatability, and reproducibility.

Validation Characteristics	Requirement		Caffeine	Riboflavin	Ascorbic Acid	Nicotinic Acid
Accuracy (recovery)	Mean	95.0–105.0%	99.6%	100.5%	99.6%	100.2%
	RSD (n = 6)	≤2.0%	0.5%	0.8%	0.4%	0.8%
Specificity	Positive control	No limits	99.2%	100.5%	101.2%	98.8%
	Negative control	≤2.0%	0.0%	0.0%	0.0%	0.0%
	Resolution	≥1.5	7.6 (Nicotinic Acid)	N.A.	6.8 (Caffeine)	4.4 (Riboflavin)
Linearity	Regression coefficient	>0.9900	0.9996	0.9995	0.9993	0.9984
Precision Repeatability	RSD (n = 6)	≤1.6%	0.3%	0.3%	0.5%	0.5%
Reproducibility	RSD (n = 6)	≤3.2%	0.8%	0.8%	2.0%	1.8%

**Table 7 pharmaceutics-15-01354-t007:** Particle size distribution profiles of the three vitamins and excipients. The X10, X50, and X90 represent the 10%, 50%, and 90% of the particles that are smaller than the corresponding particle size. The span shows the width of size distribution.

	X10 (µm)	X50 (µm)	X90 (µm)	Span ^1^
Riboflavine DSM	2.8	52.3	127.9	2.4
Nicotinic acid DSM	3.6	15.7	52.2	3.6
Ascorbic acid DSM (fine powder)	13.7	78.6	159.3	1.9
Caffeïne	3.7	27.7	140.4	4.9
Magnesium stearate	1.3	5.3	20.9	3.7
Silicon dioxide	2.0	9.6	68.1	6.9

^1^ Span = (X90 − X10)/X50.

**Table 8 pharmaceutics-15-01354-t008:** Powder mixture performance of the three different formulations, homogeneity of the three different formulations, as determined by the number of white particles at the powder surface after tapping.

	Bulk Density (g/mL)	Bulk Density (% Compared to F2)	Tapped Density (g/mL)	Tapped Density (% Compared to F2)	Hausner Ratio	Angle of Repose	Flow Properties Hausner Ratio ^1^	Flow PropertiesAngle of Repose ^2^	Homogeinity (10 Taps)	Homogeneity (20 Taps)
F1	0.686	103%	0.803	104%	1.17	35.6	Good	Fair	None	Some
F2	0.666	100%	0.771	100%	1.16	34.2	Good	Good	Many	Many
F3	0.675	101%	0.781	101%	1.16	34.6	Good	Good	Some	Some

^1^ Hausner ratio: 1.00–1.11 = Excellent, 1.12–1.18 = Good, 1.19–1.25 = Fair, 1.26–1.34 = Passable, >1.35 = Poor. ^2^ Angle of repose: 25°–30° = Excellent, 31°–35° = Good, 36°–40° = Fair, 41°–45° = Passable, 46°–55° = Poor.

**Table 9 pharmaceutics-15-01354-t009:** Results of process validation of the verum capsules. For the validation of the mixing process the average content of riboflavin, nicotinic acid and caffeine is given at the top, middle, and bottom of the powder mixture. For the validation of the filling process the average content and the acceptance value of riboflavin, nicotinic acid, and caffeine is given in the capsules. Where applicable the mean and standard deviation are given (mean ± SD).

Parameter	Riboflavin	Nicotinic Acid	Caffeine
Mixing process			
T (%)	118.9 (±7.2)	120.3 (±4.3)	99.1 (±0.8)
M (%)	110.8 (±0.4)	112.5 (±0.1)	98.9 (±0.8)
B (%)	112.5 (±2.3)	112.6 (±2.9)	98.7 (±0.1)
Capsule filling process			
Average content (%)	106.2	109.1	95.1
Acceptance value (AV)	3.5	5.4	7.9

T = top of the mixture, M = middle of the mixture, B = bottom of the mixture.

**Table 10 pharmaceutics-15-01354-t010:** Product characteristics of the uncoated capsules per batch. The parameters that are measured are the average content, uniformity of dosage unit, disintegration, and dissolution. Where applicable the mean and standard deviation are given (mean ± SD).

	Release							Stability Study		
Parameter	Specification Release	Batch						Specification End of Shelf Life	Batch	
		VB.1	VB.2	CB.V1	CB.P1	CB.V2	CB.P2		CB.V1 (6 Months)	CB.P1 (6 Months)
Assay and uniformity of dosage unit caffeine	90.0–110.0%	98.5%	102.3%	95.1%	101.4%	95.0%	99.8%	For information only	95.5%	101.4%
	AV ≤ 15.0	3.1	4.3	7.9	3.2	6.1	3.1		6.0	4.3
Assay and uniformity of dosage units riboflavin	90.0–120.0%	N.A.	N.A.	106.2%	N.A.	104.7%	N.A.	80.0–120.0% (verum)	105.8	N.A.
	AV ≤ 15.0			3.5		2.0			2.8	
Assay and uniformity of dosage units ascorbic acid	90.0–120.0%			103.7%		98.0%			104.9%	
	AV ≤ 15.0			1.9		3.1			1.9	
Assay and uniformity of dosage units nicotinic acid	90.0–120.0%			109.1%		107.3%			108.8%	
	AV ≤ 15.0			5.4		3.5			1.6	
Disintegration	NMT 15 min	<15 min	<15 min	<15 min	<15 min	<15 min	<15 min	NMT 15 min	<15 min	<15 min
Dissolution ^1^	NLT 75% in 1 h	97.9 (±2.00)%	102.1 (±1.94)%	105.1 (±1.75)%	99.1 (±2.67)%	103.9 (±0.81)%	N.A.	NLT 75% in 1 h	104.2% (±1.60)	102.7% (±1.51)

VB: Validation batch (composition of placebo capsules), CB.V: Clinical batch verum, CB.P: Clinical batch placebo. ^1^ Dissolution of the verum capsules is measured on riboflavin as index vitamin and the dissolution of the placebo capsules is measured on caffeine.

**Table 11 pharmaceutics-15-01354-t011:** The product and process characteristics of the capsules of the lab scale coater and ProCept Pan Coater (a perforated drum coater). The lag time, pulse time, R_360 min_ and the coat thickness are parameters for the product and process characteristics. The lag time is the time at which the cumulative percentage of caffeine released is 5%. The pulse time is the time at which the cumulative percentage of caffeine release is 70%. R_360 min_ is the cumulative percentage of release of caffeine at the end of the dissolution. Where applicable the mean and standard deviation are given (mean ± SD).

Parameter	Specification	Lab Scale Coater	Procept Pan Coater
Total capsules coated	n.a.	30	4500
Coating duration (hours)	n.a.	0.5–1	4
Drum speed (rpm)	n.a.	35	15
Product temperature (°C)	n.a.	30	25–30
Nozzle gas (bar)	n.a.	Not measured	0.49
Pattern gas (bar)	n.a.	Not measured	0.49
Pump speed (g/min)	n.a.	2.5–4.5	11.1–13.3
Lag time (min)	240–270	249	252
Pulse time (min)	<60	39	27
R_360 min_ (%)	>80	92 (±3.1)	106 (±7.7)
Theoretical coat thickness (mg/cm^2^)	n. a.	9.0–11.0	7.0–8.5

**Table 12 pharmaceutics-15-01354-t012:** Process and product characteristics of the coated capsules per batch. In this table the batch size, theoretical coat thickness, amount of coat liquid, the lag time, pulse time, and cumulative release are described per produced batch. Where applicable the mean and standard deviation are given (mean ± SD).

	Release										Stability Study	
Parameter	Specification	Batch										
		VB.1	VB.2	CB.V1	CB.P1	CB.V2.1	CB.P2.1	CB.V2.2	CB.P2.2	CB.V1 (6 Months)		CB.P1 (6 Months)
Batch size	For information	4500	3000	4500	4500	3140	3152	3368	3180	N.A.		N.A.
Theoretical coat thickness	7.0–8.5 mg/cm^2^	7.3 (±1.09)	7.6 (±1.50)	8.2 (±0.82)	7.4 (±1.12)	7.2 (±0.70)	7.6 (±1.20)	7.4 (±1.61)	7.3 (±0.83)			
Coat liquid	Amount of coat liquid used in gram for information	1697.9	1130.8	1699.3	1772.5	1175.3	1278	1249.3	1275.2			
Lag time	Concentration caffeine < 5% at 230 min	0.1 (±0.05)	0.1 (±0.06)	0.0 (±0.00)	0.0 (±0.02)	1.8 (±4.04)	0.1 (±0.05)	0.0 (±0.09)	0.1 (±0.12)	0.1 (±0.04)		0.2 (±0.19)
Pulse time	Concentration caffeine > 70% at 300 min	97.2 (±2.67)	103.3 (±1.96)	92.9 (±5.09)	102.2 (±2.08)	95.7 (±1.06)	103.4 (±1.55)	95.6 (±0.53)	101.0 (±1.44)	96.1 (±2.32)		100.7 (±1.42)
	T70% (min)	280	278	283	278	279	277	281	280	281		278
Cumulative release	Concentration caffeine at end of test > 80%	98.1 (±2.07)	103.4 (±2.14)	95.5 (±1.46)	102.2 (±2.11)	98.8 (±1.34)	103.9 (±0.97)	95.9 (±0.60)	101.1 (±1.34)	96.6 (±1.55)		101.7 (±1.01)

VB: Validation batch (composition of placebo capsules), CB.V: Clinical batch verum, CB.P: Clinical batch placebo. CB.V2.1/CB.V2.2 = uncoated capsules from CB.V2 are divided into two batches before coating (same goes for CB.P2.1/CB.P2.2).

## Data Availability

The dataset(s) used for the current study are available from the corresponding author upon reasonable request.

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
