# Peer review of "Capsules with Ileocolonic-Targeted Release of Vitamin B2, B3, and C (ColoVit) Intended for Optimization of Gut Health: Development and Validation of the Production Process"

_pharmaceutics, 2023, doi:10.3390/pharmaceutics15051354_

Round 1

Reviewer 1 Report

  This article "Capsules with ileocolonic-targeted release of vitamin B2, B3, and C (ColoVit) intended for optimization of gut health: development and validation of the production process" report a stable oral dosage form containing a combination of vitamin B2, B3, 387and C, intended to target the gut microbiome. The preparation and validation of this article is written in more detail. I think it can be accepted by minor revision.

  1. Line 128, it should be table 1.

  2. Figure 4, for the release of caffeine, there is only four points, it's too less.

Author Response

Reviewer #1

This article "Capsules with ileocolonic-targeted release of vitamin B2, B3, and C (ColoVit) intended for optimization of gut health: development and validation of the production process" report a stable oral dosage form containing a combination of vitamin B2, B3, and C, intended to target the gut microbiome. The preparation and validation of this article is written in more detail. I think it can be accepted by minor revision.

Authors reply: First of all, we would like to express our gratitude and appreciation to the reviewer for the thorough assessment of our manuscript and are happy to read that it has generally been well received. We have adjusted the manuscript based on the raised suggestions. In our replies below, we will elaborate on the points raised by the reviewer.

  1. Line 128, it should be table 1.

We thank the reviewer for their attentiveness and have adjusted the manuscript accordingly.

  1. Figure 4, for the release of caffeine, there is only four points, it's too less

We thank the reviewer for the comment. Based on this comment and comments raised by another reviewer, we have decided to remove figure 4 from the manuscript, because it doesn’t present any new information. The release profile of the coated capsules is further elaborated in figure 3 and table 12.

Reviewer 2 Report

Reviewer's comments for manuscript, pharmaceutics

Recommendation: Minor revisions

The authors provide a research paper on a study with a structured, validated and reproducible process to produce capsules which show a vitamin release in the ileocolonic region. The paper describes the upscaling process of the coating of the capsules for a manufacturing process and further investigates the in vitro drug release. I think the paper is very structured and thorough, but have some unclarified sections. Therefore, I suggest that the manuscript is re-considered for publications once the minor corrections have been considered and corrected.  

Please consider and address the following comments in particular: 

1.       The sentence in line 58 is very unclear, please explain in more details this higher organisms.

2.       Line 61: How is it possible to target the gut microbiota? Is it the target to the colon which is meant?

3.       I suggest moving Fig. 1 to supplementary information – seems unnecessary in the manuscript.

4.       Fig.2: Very difficult to see the scale bar and the text related to it, please improve this.

5.       I’m in doubt of the novelty of this paper. Please state this very clearly in the manuscript. And also state the perspective and how this technique can be used by others and for which purpose.

6.       In general, the captions throughout the paper should be improved to create more details on what is seen in the figure and/or table. Also if it’s mean +- SD etc.

7.       I will suggest merging Fig. 3 and 4.  

8.       There is a high inter- and intra-variability in the pH values in the GI tract of humans (especially the stomach and small intestine in healthy and diseased state). How will this coating be able to circumvent this. Please add to the paper.   

9.       What is the reason for measuring on the released caffeine and not on the actual vitamins? Do you have verified that it will be released at the same time? It is also often seen that there is interactions with the capsule or coating material. Could there be such interactions with the vitamins and the material?  

Author Response

Reviewer #2
The authors provide a research paper on a study with a structured, validated and reproducible process to produce capsules which show a vitamin release in the ileocolonic region. The paper describes the upscaling process of the coating of the capsules for a manufacturing process and further investigates the in vitro drug release. I think the paper is very structured and thorough, but have some unclarified sections. Therefore, I suggest that the manuscript is re-considered for publications once the minor corrections have been considered and corrected. 

Authors reply: First of all, we would like to express our gratitude and appreciation to the reviewer for the thorough assessment of our manuscript and are happy to read that it has generally been well received. This motivated us to further improve the manuscript based on the raised comments and suggestions. In our replies below, we will elaborate on the points raised by the reviewer.

  1. The sentence in line 58 is very unclear, please explain in more details this higher organisms.

Authors reply: We thank the reviewer for the comment. ‘Higher organisms’ in this sentence refers to humans and animals, for clearance we have restructured this sentence (Line 52-54):

Vitamins are essential micronutrients for humans and are involved in numerous biological processes. Vitamins support the immune system and function as essential precursors for enzymes involved in cellular energy generation’

  1. Line 61: How is it possible to target the gut microbiota? Is it the target to the colon which is meant?

Authors reply: We thank the reviewer for the comment. We aim to establish a controlled exposure of vitamin B2, B3, and C to the commensal bacteria residing in the ileocolonic regions of humans. Thus, we aim to ‘target gut microbiotia’ through site-specific delivery of capsule content in the terminal ileum and colon. To clarify this targeted approach, we have rephrased the sentence in line 61:

Supplementation with vitamins delivered to the gut microbiota residing in the ileocolonic regions’

  1. I suggest moving Fig. 1 to supplementary information – seems unnecessary in the manuscript.

Authors reply: We thank the reviewer for their advice and have moved the figure to the supplementary files.

  1. 2: Very difficult to see the scale bar and the text related to it, please improve this.

Authors reply: We thank the reviewer for the comment. We have improved the picture and we have also added a more descriptive caption to understand the scale bar and the text related to it.

  1. I’m in doubt of the novelty of this paper. Please state this very clearly in the manuscript. And also state the perspective and how this technique can be used by others and for which purpose.

Authors reply: To our knowledge this is the first manuscript to describe an ileocolonic-targeted vitamin capsule and its production and validation process. A study in healthy volunteers investigating the effects of colon-targeted vitamins, employing an oral colonic drug delivery system, has been published (Pham et al, gut microbes 2022). However, a validated and reproducible method intended for development and large-scale production of (ileo)colonic targeted vitamin capsules has not been described in literature. While this manuscript focusses on vitamin B2, B3, and C, the coating process as described in this manuscript can be applied to capsules containing different vitamins and vitamin mixtures. In the manuscript we describe the instruments, facilities, techniques, and validation approaches necessary to develop and produce ileocolonic-targeted capsules according to a clinical trial GMP license.

The pH-sensitive coating technique (ColoPulse) has previously been described by our research group, but for the first time, we described the transformation and implementation of the ColoPulse-coating process from small scale university laboratory facilities to a larger scale GMP-licensed production facility. Importantly, there is a demand for ileocolonic-targeted formulations with optimized release profiles to reliably deliver pharmaceutical compounds or supplements to the target location, as commercially available pH-sensitive coatings have demonstrated suboptimal release patterns under certain conditions. (please refer to line 66-84 of the introduction section of this manuscript for the rationale behind optimization of existing ileocolonic targeted strategies)

To underline the novelty of this paper, we have added the following lines to the discussion section of the manuscript: (Line 533-542)

To our knowledge this is the first manuscript to describe a validated and reproducible method to develop and produce ileocolonic-targeted vitamin capsules. There is a demand for ileocolonic-targeted formulations with optimized release profiles to reliably deliver pharmaceutical compounds or supplements to the target location, a release profile which would ideally not be affected by an inflammatory state of the intestines or inter- and intraindividual differences of intestinal physiology and function. The ColoVit utilizes an optimized pH-sensitive delivery strategy, and the large-scale production of this formulation has been validated and has proven reproducible.

 Additionally, this manuscript serves as a tech transfer of the coating process from lab scale to larger scale production of clinical batches in a GMP licensed facility.

  1. In general, the captions throughout the paper should be improved to create more details on what is seen in the figure and/or table. Also if it’s mean +- SD etc.

Authors reply: We thank the reviewer for the comment. We have improved the captions for figure 1 and 3, table 9-12, provided more details related to the information shown in the figures and table.  

  1. I will suggest merging Fig. 3 and 4.  

Authors reply: We thank the reviewer for the comment. Based on this comment and comments raised by another reviewer, we have decided to remove figure 4 from the manuscript, because it doesn’t present any new information. The release profile of the coated capsules is further elaborated in figure 3 and table 12.

  1. There is a high inter- and intra-variability in the pH values in the GI tract of humans (especially the stomach and small intestine in healthy and diseased state). How will this coating be able to circumvent this. Please add to the paper. 

Authors reply: We thank the reviewer for this important comment, as the inter- and intra-variability of intestinal pH, especially in an inflammatory/diseased state, can highly influence release characteristics of oral (ileo)colonic drug delivery systems. Within our research group, we have published several papers on pre-clinical and clinical studies regarding the performance of the ColoPulse coating. (1-3)

In line 85-116 of the introduction, we now highlight the manner through which the described coating will circumvent the variability in pH and transit-time, and how the ColoPulse technology will realize optimized release profiles.

The ColoPulse leverages a super disintegrant (e.g. Ac-Di-Sol®) incorporated into a pH-sensitive polymer (e.g. Eudragit® S) in a non-percolating fashion. The outer layer of the ColoPulse starts to dissolve once an intraluminal pH >7 is reached, and the super disintegrant enables drug release within 15 minutes. Importantly, the peak in pH >7 existing in the terminal ileum appears to be highly consistentunder both normal physiological- and disease/inflammatory conditions. Gastrointestinal pH proximal to the terminal ileum does not exceed a pH threshold of 7. Colonic transit time can be increased in a diseased state, and commercially available pH-sensitive colonic release systems have demonstrated insufficient release characteristics under these conditions, however, the ColoPulse enables timely release through the incorporation of super disintegrants. Importantly, we have validated the ColoPulse coating technology in both healthy volunteers and patients with Crohn’s disease, thus patients with diseased intestines. The ColoPulse was found to reliable release its contents in both the healthy volunteer and the CD group.

  1. Maurer MJ, Schellekens RC, Wutzke KD, Dijkstra G, Woerdenbag HJ, Frijlink HW, Kosterink JG, Stellaard F. A non-invasive, low-cost study design to determine the release profile of colon drug delivery systems: a feasibility study. Pharm Res. 2012 Aug;29(8):2070-8. doi: 10.1007/s11095-012-0735-3. Epub 2012 Mar 16.
  2. Maurer JM, Schellekens RC, van Rieke HM, Wanke C, Iordanov V, Stellaard F, Wutzke KD, Dijkstra G, van der Zee M, Woerdenbag HJ, Frijlink HW, Kosterink JG. Gastrointestinal pH and Transit Time Profiling in Healthy Volunteers Using the IntelliCap System Confirms Ileo-Colonic Release of ColoPulse Tablets. PLoS One. 2015 Jul 15;10(7):e0129076. doi: 10.1371/journal.pone.0129076. eCollection 2015.
  3. Maurer JM, Schellekens RC, van Rieke HM, Stellaard F, Wutzke KD, Buurman DJ, Dijkstra G, Woerdenbag HJ, Frijlink HW, Kosterink JG. ColoPulse tablets perform comparably in healthy volunteers and Crohn's patients and show no influence of food and time of food intake on bioavailability. J Control Release. 2013 Dec 28;172(3):618-24. doi: 10.1016/j.jconrel.2013.09.021. Epub 2013 Oct 2.

  1. What is the reason for measuring on the released caffeine and not on the actual vitamins? Do you have verified that it will be released at the same time? It is also often seen that there is interactions with the capsule or coating material. Could there be such interactions with the vitamins and the material?  

Authors reply: We thank the reviewer for the comment. We have decided to measure the released caffeine as a marker, because the vitamins are not chemically stable in the different dissolution media of the gastro-intestinal simulation system (GISS). We have also added this information to the discussion (line 478-483):

Additionally, 5.9% of caffeine is added to the capsules as a marker substance for the drug release. Caffeine is added as a marker substance, because compared to the vitamins it is stable in the different dissolution media of the gastro-intestinal simulation system (GISS) and it has similar dissolution properties.

We have measured the dissolution of the uncoated capsules on riboflavin as an index vitamin to confirm that the vitamins are also released. During stability testing, we have not seen any indication of an interaction between the vitamins and the coating material.

Reviewer 3 Report

The authors submitted a study on the validation of the manufacturing process of a vitamin loaded, colon targeting capsule. Part of the study was also a tech transfer of the coating from a very limited lab scale to a small clinical batch scale. The results show that the capsules fulfill the pharmacopeial and in-house requirements. The study is worth publishing but some major issues have to be addressed before acceptance is possible.

The most pressing matter is the lacking discussion of the results. The first two paragraphs belong in to the introduction of the manuscript. They give background information but are not part of the results discussion. The next two paragraphs actually discuss results, followed by two paragraphs on background information about the actual mechanism of action of the included active substances also not covering results. Please only discuss the results of the experiments in the discussion section. Please make connections between your experiments and draw conclusions, e.g., is it more beneficial to use PH101 or PH102 for the capsule filling? Does the flowability have an influence?

The second point that has to be improved is the discussion about the influence of the different coater on the coating quality. The authors use information from a publication on wire arc coating to discuss their results. Wire arc coating includes the melting of metals in an electrical arc and is not comparable to spray coating. It is highly unlikely that the mechanisms are similar. In the manuscript the authors speak of the impact velocity of particles, which is true for wire arc coating applications. In the study, droplets are generated and it is unlikely that the dry completely before impact. Also, it is not clear what the authors mean with the density/porosity of the  coating. Crucial information about the lab scale coating is missing (spray rate, duration, temperature, etc.) so it is not possible to better evaluate the data. My first assumption it the formation of a more homogeneous coating in the ProCept coater compared to the lab scale batch, allowing a reduction of the coating mass. Can this be a possibility? Please speak of theoretical coating thickness unless it was measured.

Also, the method section is in some parts not conclusive. For example, SEM images were captured but the method is not described. The tech transfer section only explains in-vitro drug release but not the tech transfer. Place it in the in-vitro drug release section, this is not logical.

More comments are listed in the following.

2.2. What were the settings of the powder dispersion unit?

Table 2: Falsely labelled as Table 1. Please include the relative content also behind the absolute amount similar to Table 1.

Figure 1: What is the reason for the shoulder of ascorbic acid?

2.7: What equipment was used for the in vitro drug release? Please mention it in brief.

Table 7: One significant digit is sufficient for this kind of data and this kind of technique.

Table 8 a: The Hausner ratio is not presented in percent. Why is only the angle of repose used to qualify the flowability? Are theses results indicative for your experimental results? This is not discussed.

Table 8 b: I struggle to find meaning in this analysis. The data is highly subjective and also not discussed. Can this be quantified more? Can you include pictures that demonstrate what you mean with “some” and “many”?

Table 9: Capsule filling-uniformity of content: please add the standard deviation. The uniformity of content cannot be expressed by an average value. General comment: Add a measure for the data spread whenever possible, e.g. table 11.

Author Response

Reviewer #3

The authors submitted a study on the validation of the manufacturing process of a vitamin loaded, colon targeting capsule. Part of the study was also a tech transfer of the coating from a very limited lab scale to a small clinical batch scale. The results show that the capsules fulfill the pharmacopeial and in-house requirements. The study is worth publishing but some major issues have to be addressed before acceptance is possible.

Authors reply: First of all, we would like to express our gratitude and appreciation to the reviewer for the thorough assessment of our manuscript and are happy to read that it has generally been well received. This motivated us to further improve the manuscript based on the raised comments and suggestions. In our replies below, we will elaborate on the points raised by the reviewer.

  1. The most pressing matter is the lacking discussion of the results. The first two paragraphs belong in to the introduction of the manuscript. They give background information but are not part of the results discussion. The next two paragraphs actually discuss results, followed by two paragraphs on background information about the actual mechanism of action of the included active substances also not covering results. Please only discuss the results of the experiments in the discussion section. Please make connections between your experiments and draw conclusions, e.g., is it more beneficial to use PH101 or PH102 for the capsule filling? Does the flowability have an influence?

Authors reply: We thank the reviewer for the comment. We have restructured the manuscript as advised. We have relocated the first two paragraphs of the discussion to the introduction. Although the paragraphs regarding the mechanism of action of the included active substances do not directly address the results, we do think it is important to provide the reader with the relevance and rationale behind the chosen substance of interests and understand why an (ileo)colonic-targeted approach is considered above conventional delivery strategies. Additionally, another reviewer wanted us to further highlight for which purpose this technique and capsules could be applied. However, we agree with the reviewer that the primary discussion should be more focussed on the conducted experiments, and for this purpose we have added a more extensive discussion on the results in the discussion section (line412-432):

We started with the formulation of the uncoated capsules. The verum capsules contain 9.5% riboflavin, 0,68% nicotinic acid and 63.5% ascorbic acid per capsule. Additionally, 5.9% of caffeine is added to the capsules as a marker substance for the drug release. Caffeine is added as a marker substance, because compared to the vitamins it is stable in the different dissolution media of the gastro-intestinal simulation system (GISS).

Combining four different compounds in a single dose brings different challenges. The difference in particle size, particle morphology, density and quantity of the four compounds are critical for the homogeneity of the powder mixtures and the content uniformity of the capsules. To successfully combine the compounds in a single dose we have characterized all the raw materials and formulated three different powder mixtures. The powder mixture with Avicel®PH101 (formulation 1) had a slightly poorer flowability than the powder mixtures with Avicel®PH102(formulation 2) and a mixture of Avicel®PH101: Avi-cel®PH102 (50:50) (formulation 3). However, the visual homogeneity of formulation 1 was better than the other two formulations. This can be explained by the particle size of Avicel®PH101; Avicel®PH101 has a smaller nominal particle size than Avicel®PH102. The smaller nominal particle size fits better with the different particle sizes of the four compounds, which results in a more stable formulation. On top of that, the slightly poorer flowability also results in a more stable formulation without causing problems during the filling process.

  1. The second point that has to be improved is the discussion about the influence of the different coater on the coating quality. The authors use information from a publication on wire arc coating to discuss their results. Wire arc coating includes the melting of metals in an electrical arc and is not comparable to spray coating. It is highly unlikely that the mechanisms are similar. In the manuscript the authors speak of the impact velocity of particles, which is true for wire arc coating applications. In the study, droplets are generated and it is unlikely that the dry completely before impact. Also, it is not clear what the authors mean with the density/porosity of the  coating. Crucial information about the lab scale coating is missing (spray rate, duration, temperature, etc.) so it is not possible to better evaluate the data. My first assumption it the formation of a more homogeneous coating in the ProCept coater compared to the lab scale batch, allowing a reduction of the coating mass. Can this be a possibility? Please speak of theoretical coating thickness unless it was measured.

Authors reply: We thank the reviewer for this important comment. We have removed the information from the publication of the wire arc coating and have adjusted the discussion (line 440-463):

After the development and validation of the uncoated capsules, the development and the validation of the coating process started. The coating process was transferred from a lab scale process (30 units, coating duration of approximately 30 – 60 min) to a controlled production scale process (4500 units, coating duration of approximately 4 hours ) [32, 33]. Unfortunately, we were not able to control and measure all the coating parameters of the lab scale coater. The only parameters that could be measured were the drum speed, the pump speed and the temperature. Experience with the lab coater has shown that during the coat process the temperature of the product has to be between 25 and 30 °C to obtain the desired properties of the coating. Variation in the drum and pump speed only influ-enced the total coating duration, but it did not affect the properties of the coating. Howev-er, the pump speed was set up to the highest possible speed on the pump to prevent clog-ging.                                                                                                                                                                              Interestingly the different equipment and process parameters allowed us to apply a 25% thinner coating (8 vs. 10 mg/cm2) while still obtaining similar release properties. This difference is explained by a higher density of the coating as produced at production scale. The coating density is the volume of coating material used to cover the area of the cap-sules. The reasons for this are the higher forces exerted on the coating due to the increased mass and the longer duration of the process. Additionally, we were able to control the nozzle gas which was not possible at lab scale. Increasing the nozzle gas results in the formation of smaller particles, this eventually leads to a higher coating density (and re-duced porosity of the coating). The porosity of the coating can be described by the area of the capsule that is not covered with the coating: in other words, the holes in the coating. This explains the similarity between the release profiles of products from the different processes despite the difference in theoretical coat thickness.

The difference between the theoretical coating thickness could not be explained with formation of a more homogeneous coating in the ProCept coater compared to the lab scale batch, allowing a reduction of the coating mass. This is due to the fact that in both cases the coating is made separately in erlenmeyer flask and is stirred constantly with a magnetic stirrer. In both cases the coating is pumped from the flask to the coating drum. We don’t expect any differences between the homogeneity of the coating solution of the ProCept coater compared to the lab scale coater. 

  1. Also, the method section is in some parts not conclusive. For example, SEM images were captured but the method is not described. The tech transfer section only explains in-vitro drug release but not the tech transfer. Place it in the in-vitro drug release section, this is not logical.

Authors reply: We thank the reviewer for the comment. Particle morphology was visualized by scanning electron microscopy (JEOL JSM-6301-F, Jeol, Japan) with a magnification factor of 50. This is described in paragraph 2.2 (line 144-145). We have also adjusted paragraph 3.4 and added new information to table 11 to focus more on the tech transfer.

  1. What were the settings of the powder dispersion unit?

Authors reply: We thank the reviewer for the comment. There are no specific settings for the capsulating machine, the machine uses the force of gravity of fill the capsules. We have added the target fill weight to clarify the process more. Line 144 – 147: Capsules were filled on an automated filling capsulating machine (INCAP, Dott. Bona-pace & C) operated at a rate between 1500 and 3000 capsules per hour with a target fill weight of 425 mg.

  1. Table 2: Falsely labelled as Table 1. Please include the relative content also behind the absolute amount similar to Table 1.

Authors reply: We thank the reviewer for their attentiveness and have adjusted the manuscript accordingly.

  1. Figure 1: What is the reason for the shoulder of ascorbic acid?

Authors reply: We thank the reviewer for the comment. The shoulder in the chromatogram for ascorbic acid is visible, because for the determination of caffeine, riboflavin, and nicotinic acid a different dilution of the sample is injected than for the determination of ascorbic acid. The shoulder is due to the fact that the concentration of ascorbic acid is too high, the sample for the assay of ascorbic acid is diluted first and injected afterwards. While the sample for caffeine, riboflavin, and nicotinic acid is not diluted. In this case the injection of caffeine, riboflavin and nicotinic acid is shown. The shoulder of ascorbic acid is not visible in the diluted sample. The chromatogram has also been moved the supplementary information (on the recommendation of reviewer 2)

  1. 7: What equipment was used for the in vitro drug release? Please mention it in brief.

Authors reply: We thank the reviewer for the comment. We have described the method and the equipment in more detail in line 260 – 270. For this dissolution test the pharmacopoeial paddle method is used, the formulation is exposed to four phases simulating in subsequent order the stomach, the jejunum, distal ileum, and the proximal colon. In the article : Schellekens, R.C.A.; Stuurman, F.E.; Van der Weert, F.H.J.; Kosterink, J.G.W.; Frijlink, HW. A novel dissolution method relevant to intestinal behaviour and its application in the evaluation of modified release mesalazine products. Eur J Pharm Sc. 2007, 30, 15-20 doi: 10.1016/j.ejps.2006.09.004 the composition of each phase is described in more detail. Reference is made to this article.

  1. Table 7: One significant digit is sufficient for this kind of data and this kind of technique

Authors reply: We thank the reviewer for their attentiveness and have adjusted the manuscript accordingly.

  1. Table 8 a: The Hausner ratio is not presented in percent. Why is only the angle of repose used to qualify the flowability? Are theses results indicative for your experimental results? This is not discussed.

Authors reply: We thank the reviewer for their attentiveness and have adjusted the Hausner ratio. We have also taken the Hausner ratio in account to determine the flowability and added this data to table 8A. The discussion is adjusted, see comment 1.

  1. Table 8 b: I struggle to find meaning in this analysis. The data is highly subjective and also not discussed. Can this be quantified more? Can you include pictures that demonstrate what you mean with “some” and “many”?

Authors reply: We thank the reviewer for the comment. We have added Figure 2 to the manuscript with the visualization of the homogeneity based on the number of white particles at the powder surface.

  1. Table 9: Capsule filling-uniformity of content: please add the standard deviation. The uniformity of content cannot be expressed by an average value. General comment: Add a measure for the data spread whenever possible, e.g. table 11.

Authors reply: We thank the reviewer for the comment. We have adjusted table 9, the content uniformity and uniformity of dosage unites is described by the Acceptance value, the average content is only given as extra information. We have also added standard deviations to table 11 as a measure for the data spread.

Round 2

Reviewer 2 Report

No further comments. I think the authors corrected the manuscript accordingly to the advise hence, I suggest the manuscript to the accepted for publication. 

Author Response

Thank you for your time

Reviewer 3 Report

The authors have answered all but one of my questions sufficiently because of a misunderstanding of the comment. As this comment deals with an important processing step, the coating, I would like to rephrase my comment to make clear what I meant.

In my second comment, I wrote that I do not think that the coating porosity is necessarily the (only) defining factor that allows a reduction of the coating mass, but rather the homogeneity of the coating. The authors connected my comment to the homogeneity of the coating solution. What I meant, however, is the homogeneity of the applied coating. As the authors state, it was possible to generate a finer spray with the ProCept coater and to operate a generally more stable process.

A likely outcome of the more stable process in the ProCept coater is a more homogeneous coverage of the particles with the coating solution. Meaning that with less material, you will obtain already a complete and homogeneous coating, also a result of polymer coalescence during the prolonged processing time at increase temperature. Conversely, with the poorly controlled lab scale coater operated with a coarser spray over a shorter time, coverage of your particle will be less controlled. This would very likely resulting in larger polymer droplets on the particle surface that do not readily coalesce, thereby requiring a higher coating solution mass until complete and homogeneous coverage is achieved.

May it be possible that such a phenomenon also played a role in your process?

Author Response

We thank the reviewer for the comment and the explanation, we apologize for the misunderstanding. The described phenomenon definitely played a role in our process, the higher coating density that we describe in the discussion is also partially caused by the phenomenon described by the reviewer. We have adjusted the discussion and have added the explanation of the  more homogenous coverage of the coating to line 531 -537:   The prolonged processing time, the formation of smaller particles, and the stability of the coat process with the ProCept coater also lead to a more homogenous coverage of the capsules with the coating. Due to the prolonged processing time, the polymer particles have more time to coalesce, which results in a more homogenous coating. Conversely, with the lab scale coater a higher coating solution mass is required to achieve the same homogeneous coverage.